Earth System Science Data Discussions — Open Access

# Twenty-one years of hydrological data acquisition in the Mediterranean Sea: quality, availability, and research

Alberto Ribotti[1], Roberto Sorgente[1], Federica Pessini[1], Andrea Cucco[1], Giovanni Quattrocchi[1], Mireno Borghini[2]

[1]Istituto per lo studio degli impatti Antropici e Sostenibilità in ambiente marino (IAS) of CNR, 09170 Oristano, Italy, https://orcid.org/0000-0002-6709-1600, https://orcid.org/0000-0003-0268-7822, https://orcid.org/0000-0002-1744-9117, https://orcid.org/0000-0002-4469-2286, https://orcid.org/0000-0002-0020-1780
[2]Istituto di Scienze Marine (ISMAR) of CNR, 19032 La Spezia, Italy, https://orcid.org/0000-0002-5654-4731

*Correspondence to*: Federica Pessini (federica.pessini@ias.cnr.it)

**Abstract.** Since 2000 and for the following 20 years hydrological data of the Mediterranean Sea, with a particular focus on the western and central Mediterranean sub-basins, have been acquired to study the hydrodynamics at both coastal and open sea scales. Totally 1468 hydrological casts were realized in 29 oceanographic cruises planned due to scientific purposes linked with funding research projects but sometimes driven by sea conditions and type of vessel. After an accurate quality assurance and control, following standard procedures, all hydrological data were included in four online public open access repositories in SEANOE available from https://doi.org/10.17882/87567 (Ribotti et al., 2022). Hydrological and dissolved oxygen data are always present in all the datasets whereas pH, fluorescence, turbidity and CDOM are available just for some cruises. Samplings were carried out mainly along transects with some repetitions in the years. Then the results of two data analyses, staircase systems in the Tyrrhenian Sea and in the Algero-Provencal sub-basin and spreading of the Western Mediterranean Transient, are mentioned.

**Keywords:** Mediterranean, CTD, hydrological data, circulation, climatology

## 1 Introduction

The Mediterranean basin is an east-west oriented semi-enclosed sea connected with the North Atlantic Ocean, through the narrow and shallow Gibraltar Strait. Its circulation, although characterized by complex patterns, can be simply described as made of three main active overturning circulation cells, a shallow and two deep ones (Tanhua et al. 2013). The Mediterranean Sea is one of the few basins in the world where deep convection and water mass formation take place (Siedler et al., 2001, page 422). Morphology, climatology, and hydrodynamics, with also reduced timescales in the turnover of most of the oceanographic processes of the global ocean, make the Mediterranean Sea a "small and accessible ocean" where changes can be more easily studied (Bethoux et al., 1999).

Within this context, the Italian National Research Council (CNR), organized over 29 scientific cruises in 21 years in the Mediterranean, mostly in the western Mediterranean and the Sicily Channel, with a particular focus on the acquisition of vertical profiles of temperature, salinity, dissolved oxygen, fluorescence and, more recently, pH and turbidity. The cruises were always planned on funding projects' objectives but then adding further activities linked to other research interests and/or scientific collaborations. Therefore, for each cruise and dataset there is a duality between 'fit for purpose', i.e. the projects that prompted us to organize the campaigns, and 'fit for use', i.e. the possibility of reusing the data for purposes other than the original aims, i.e. data also acquired in areas outside projects' interest. This duality was also behind the sampling plan that could be on a regular or quite regular grid (cruises 2000-2004 and 2014), generally



suitable to calibrate and validate ocean numerical models, and/or along transects (after 2004) for studies on general or local circulation and, if with some repetition in the years, for climatological studies also (for example Gibraltar-Sardinia, Balears-Sardinia, Sardinia-Tunisia) and finally, sometimes linked with the maintenance of the instruments

installed on deep moored chains like in Tuscany-Corsica and Sicily-Tunisia. Some cruises or parts of them also show any apparent regular spatial resolution like during a cruise in 2012 with the transect Balears-Sardinia and apparently scattered casts in south Tyrrhenian-Sardinia Channel due to different scientific purposes.

Regardless by any sampling plan, the data acquired over the 2000-2020 period allowed us to characterize the different water masses (Ribotti et al., 2004; Puillat et al., 2006; Santinelli et al., 2008; Belgacem et al., 2020) and the ocean

circulation mainly, but not only, in the seas around the Sardinia Island (Sorgente et al., 2016; Pessini et al., 2018, 2020), and to monitor changes with time along repeated transects (Schroeder et al., 2008) or single casts (Durante et al., 2020). But also to calibrate and validate ocean numerical models at different spatial scales, from open ocean to coastal (Sorgente et al. 2003), also for issues of marine environmental management (oil spill combat; Gonnelli et al., 2016; Sorgente et al. 2016).

The paper is organized as follow: in the methods, cruises, vessels, and data are described, as well as the quality assurance and check procedures applied to the datasets, subsequently two examples of their use: the results obtained from climatological analyses on the diffusion of the Western Mediterranean Transient and a still open issue on thermohaline staircases in the Western Mediterranean are described.

**2. Methods**

Over 29 oceanographic cruises were realized in the Mediterranean basin between May 2000 and October 2020 (table 1 and figure 1) by the CNR in Oristano jointly with the CNR in Spezia, except for the years 2016 and 2019 when no cruises were carried out. The strict collaboration, since 2000 till 2018, with the CNR institute in La Spezia, with a much longer experience in ocean cruises and studies, has brought to the choice of some sampling plans like the transects in the Sicily and Corsica channels where they moored deep oceanographic chains for tens of years.

Totally, 1468 hydrological vertical profiles were acquired in 21 years in the Mediterranean Sea, covering areas spanning from the Gibraltar Strait in the western subbasin to south of Crete in the eastern part of the Mediterranean Sea. In addition to projects' objectives and further research interests, two more aspects have influenced, even heavily in the last four years, data acquisition: sea conditions and type of vessel (described in the next subsection). Until 2017 (table 1), the availability of large research CNR vessels like Urania and Minerva Uno, allowed working throughout the

Mediterranean and in "not optimal" sea conditions. Whereas, since 2018 the available research vessels were not suitable to work in rough sea conditions in the open sea. This, along with the increase of research projects focused on the coastal environment and with shortening of the available times for the vessels usage, limited the 'fit for use' in the areas outside the projects' aims.

**1.1 Research vessels and projects**

Different research vessels were used during the several measuring campaigns, each one with characteristics and facilities that influenced both sampling plans and data acquisitions. Between May 2000 and December 2017, the CNR used two multidisciplinary research vessels designed in Italy that permitted to work all over the Mediterranean and in all the seasons: the 61.30 meters long R/V Urania (2000-2014) and the 47,66 meters long R/V Minerva Uno (2015-2017). During these years the performed cruises allowed to acquire data to study the local (e.g. the Sardinia Sea till



2004) and the general circulation all over the Mediterranean, to maintain deep ocean bottom-moored infrastructures in
the Corsica, Sardinia and Sicily channels and to acquire data to calibrate and validate numerical models at different
spatial scales. In 2016 no cruise was planned.

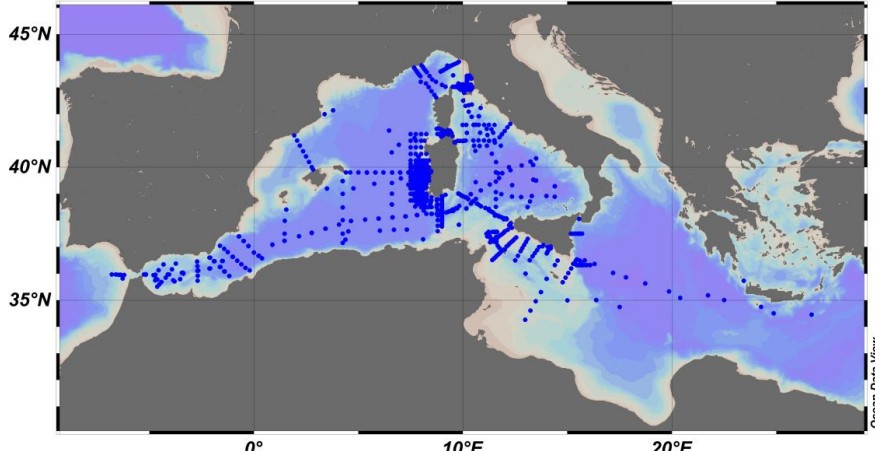

**Figure 1. All the CTD casts acquired during the 29 cruises in the Mediterranean between 2000 and 2020.**


| Cruise name | Start cruise | End cruise | # casts | Vessel | Areas | Ocean parameters |
|---|---|---|---|---|---|---|
| Medgoos1 | 28/05/2000 | 02/06/2000 | 38 | Urania | S Sardinia Sea & Channel | P, C, T, O, F |
| Medgoos2 | 23/03/2001 | 03/04/2001 | 67 | Urania | Sardinia Sea & Channel | P, C, T, O, F |
| Medgoos3 | 10/09/2001 | 20/09/2001 | 41 | Urania | S Sardinia Sea & Channel | P, C, T, O, F |
| Medgoos4 | 04/05/2002 | 23/05/2002 | 68 | Urania | Sardinia Sea & Channel | P, C, T, O, F |
| Medgoos5 | 31/10/2002 | 18/11/2002 | 53 | Urania | S Sardinia Channel, Sicily Strait | P, C, T, O, F |
| Medgoos6 | 28/03/2003 | 17/04/2003 | 92 | Urania | S Sardinia Sea & Channel, Sicily Strait, Tyrrhenian Sea | P, C, T, O, F |
| Medgoos7 | 07/01/2004 | 26/01/2004 | 87 | Urania | Sardinia Sea, Sicily Strait, Tyrrhenian Sea | P, C, T, O, F |
| Medgoos8 | 08/05/2004 | 21/05/2004 | 67 | Urania | Tyrrhenian Sea, Corsica Channel | P, C, T, O, F |
| Medgoos9 | 06/10/2004 | 23/10/2004 | 71 | Urania | Gibraltar Strait, Alboran Sea, Algerian basin, Sardinia Channel, Tyrrhenian Sea, Sicily Strait | P, C, T, O, F |
| Medgoos10 | 28/05/2005 | 10/06/2005 | 42 | Urania | Alboran Sea, Algerian basin, Sardinia Channel, Tyrrhenian Sea | P, C, T, O, F |
| Medgoos11 | 19/11/2005 | 01/12/2005 | 34 | Urania | Tyrrhenian Sea, Sicily Strait | P, C, T, O, F |
| Medgoos12 | 02/04/2006 | 17/04/2006 | 26 | Urania | Sicily Channel, Ionian Sea | P, C, T, O, F |
| Medgoos13 | 07/10/2006 | 28/10/2006 | 56 | Urania | Gibraltar Strait, Alboran Sea, Algerian basin, Sardinia & Corsica channels | P, C, T, O, F |





| | | | | | | |
|---|---|---|---|---|---|---|
| MedCO07 | 05/10/2007 | 29/10/2007 | 84 | Urania | E Ionian, Sicily Strait, Sardinia Channel, Tyrrhenian Sea, Ligurian Sea, Catalan Sea, Algerian basin | P, C, T, O, F |
| MedCO08 | 03/11/2008 | 24/11/2008 | 63 | Urania | Gibraltar Strait, Alboran Sea, Algerian basin, Sardinia & Corsica channels | P, C, T, O, F |
| Sicily09 | 30/10/2009 | 23/11/2009 | 109 | Urania | channels of Sardinia, Sicily, Corsica, Bonifacio Strait | P, C, T, O, F |
| Bonifacio2010-Cor | 12/03/2010 | 22/03/2020 | 28 | Maria Grazia | Bonifacio Strait, Corsica Channel | P, C, T, O |
| Bonifacio2010-Sic | 23/11/2010 | 09/12/2010 | 22 | Urania | Sardinia Channel, Sicily Strait | P, C, T, O |
| Bonifacio2011 | 09/11/2011 | 23/11/2011 | 19 | Urania | Algero-Provencal basin, Tyrrhenian Sea, Sardinia Channel | P, C, T, O, F, Tu |
| Ichnussa2012 | 11/01/2012 | 27/01/2012 | 36 | Urania | Algerian basin, Sardinia Channel, Tyrrhenian Sea | P, C, T, O, F |
| Ichnussa2013 | 15/10/2013 | 29/10/2013 | 44 | Urania | E Ionian, Sicily Strait, Sardinia & Corsica channels, Tyrrhenian Sea, Algerian basin | P, C, T, O, F |
| Ichnussa2014 | 13/11/2014 | 01/12/2014 | 68 | Urania | Tyrrhenian Sea, Sardinia & Corsica channels, Ligurian Sea, Algero-Provencal basin | P, C, T, O, F |
| SeriousGame2014 | 17/05/2014 | 18/05/2014 | 45 | CP 406 A. Scialoja | N Tyrrhenian Sea | P, C, T, O, F, Tu |
| | 21/05/2014 | 21/05/2014 | 45 | | | P, C, T, O, F, Tu, CD |
| Ichnussa2015 | 25/11/2015 | 14/12/2015 | 66 | Minerva Uno | Tyrrhenian Sea, Sicily Channel | P, C, T, O, F |
| Ichnussa2017 | 23/10/2017 | 09/11/2017 | 37 | Minerva Uno | channels of Sardinia, Corsica and Sicily, Sicily Strait | P, C, T, O, F |
| Piattaforme2018 | 18/05/2018 | 03/06/2018 | 29 | G. Dalllaporta | Sicily Channel, Sicily Strait | P, C, T, O, F, pH, Tu |
| Ichnussa2018 | 20/09/2018 | 03/10/2018 | 10 | G. Dalllaporta | Sicily Strait | P, C, T, O, F |
| Ichnussa2020 | 09/10/2020 | 10/10/2020 | 21 | G. Dalllaporta | N-NE Sardinia | P, C, T, O, F, Tu |

**Table 1. List of cruises carried out from 2000 to 2020. Areas with at least three casts are listed. Ocean parameters are listed with acronyms, so P stands for pressure, C for conductivity, T for temperature, O for dissolved oxygen, F for fluorescence, Tu for turbidity, CD for CDOM and pH is just pH.**

In March 2010, the 42.35 m long R/V Maria Grazia was used just for the cruise Bonifacio2010-Cor for a project in the

Bonifacio Strait area where a an oceanographic and oil spill prediction system based on numerical models was realized for the local Coast Guard.

During the years 2018-2020, the 35.7 m long R/V Gianfranco Dallaporta, a coastal vessel particularly suitable for activities related to scientific fishing and marine biology, was used in research projects mainly focused on coastal measurements and launch/rescue of Lagrangian surface drifters in south Sicily and Sardinia. In 2019 the 8-days long

cruise Ichnussa2019 was cancelled due to bad weather conditions.



In May 2014, the 21.5 meters long patrol boat CP406 Antonio Scialoja, a unit of the Italian Coast Guard, was used during a set of two daily surveys named SeriousGame2014-1 and -2 performed in the Tuscan Archipelago, northern Tyrrhenian. Data were acquired to test one of the limited ocean forecast models for the oil spill fate (Gonnelli et al., 2016; Sorgente et al., 2016), part of the wider Mediterranean decision support system (Zodiatis et al., 2016).

**1.2 Sensors and data**

For all the adopted vessels, the hydrological instruments were mounted on a General Oceanic rosette system with 24x12-l. Niskin bottles for water column samples collection (2x10-l. during the cruise Bonifacio2010-Cor). This system was equipped with a SBE911 plus CTD probe (Sea-Bird Inc.), sampling the hydrological parameters along the vertical at 24 Hz and with a lowering speed of 1 m/s. With the CTD, data of water conductivity was measured by a SBE-4
sensor, with a resolution of $3 \times 10^{-4}$ S/m; the water temperature by means of a SBE-3/F thermometer, with resolution of 0.00015 °C/bit at -1 °C or 0.00018 °C/bit at 31 °C; the dissolved oxygen (DO) by means of a SBE-13, with a resolution of 4.3 μM, from Medgoos1 to 4, then using a SBE-43 polarographic membrane sensor with a range of 120% of surface saturation and an accuracy of ±2% of saturation; the Chlorophyll-a fluorescence, reported as Relative Fluorescence Unit (Chl-a), by means of a Sea Tech Inc. fluorometer with energy emitted by the flash lamp of 0.25 J for flash, with a
temperature range between 0 ° - 25 °C and a resolution of 0.15 g/l.

Since the cruise Medgoos12, in April 2006, the Chelsea Aqua 3 fluorometer has been used, whereas, due to sensor malfunction, no fluorescence data were collected during the Medgoos11 and the two cruises Bonifacio2010-Sic and - Cor. Turbidity was acquired just during Bonifacio2011 by a Seapoint, and by a WET Labs ECO-NTU during Piattaforme2018 and Ichnussa2020. The pH/Redox sensor (SBE27) was available just during the cruise
Piattaforme2018 but calibrated one year before. During cruises from Medgoos5 to 7 temperature data were checked onboard at defined depths against inverted thermometers, model RTM 4002 by Sensoren Instrumente Systeme GmbH (SiS), installed in correspondence of the Niskin bottles number 1, 3, 5, 7 of the rosette sampler.

Since the cruise Medgoos2 in March-April 2001 (see table 1), redundant or secondary sensors were always used for a data quality assessment (as defined in Bushnell et al., 2019) for measurements of temperature, salinity, and dissolved
oxygen (this from Medgoos5 in October 2022; table 1). These secondary sensors were used to evaluate the stability of primaries ones on board and during the following visual data quality check.

Pre-cruise and post-cruise calibrations of first and redundant sensors of temperature, salinity and dissolved oxygen were performed by CNR technicians at the NATO-CMRE Center in La Spezia (Italy). These two calibrations permitted to obtain a slope correction, used in the configuration file of the SBE Seasoft™ suite of programs, improving the data
quality. The post-cruise calibration was not performed on data acquired during the cruise Piattaforme2018. After their acquisition, data were pre- processed by the SBE Data Processing™ software with the updated configuration file, to correct coarse errors. Data from the redundant sensor were used instead of the primary in case of malfunction (Ribotti et al., 2020).

But, despite their double calibrations, during a cruise all sensors can drift then reducing their data quality. For
conductivity and dissolved oxygen sensors, their stability was checked through on-board comparisons with data from water samples. On board the R/V Urania (2000-2014) and R/V Minerva Uno (2015-2017), salinity was checked against the onboard analyses of water, sampled at pre-defined depths, by a Guildline™ 8400B Autosal Laboratory Salinometer, while on the other vessels all samples were analysed once back at CNR (Ribotti et al., 2020). The same for dissolved oxygen measurements that were checked onboard against Winkler titration method (Winkler, 1888) and analysed
through the programme TIAMO 2.0™ by Metrohm.



Finally, Chl-a, pH and turbidity data were not calibrated. The pressure sensor is usually stable and so it is not calibrated but, in case of malfunction, it was sent to SBE Inc. in USA to be revised.

During all cruises also dissolved inorganic nutrients were acquired from water samples and analyses were carried out once back in the laboratory. Data, from Medgoos8 in May 2004 to Ichnussa2017 in October-November 2017, are available online (Belgacem et al., 2019) and described, in their quality check procedures, in Belgacem et al. (2020).

Furthermore, in the framework of the MEDESS4MS project (see info on https://keep.eu/), on 17 and 21st May 2014 two daily cruises, SeriousGame2014-1 and -2, were jointly held by the CNR in Oristano, the CNR in Ancona and the Italian Coast Guard. The two surveys covered, with 45 casts each, about 880 km$^2$ of shelf on a 5 x 5 km spatial grid in latitude and longitude of an area located north of the Elba Island in the Tuscany Archipelago, northern Tyrrhenian Sea. The CTD probe was a Seabird SBE19 plus V2 equipped with sensors of pressure, temperature, conductivity, SBE 63 for dissolved oxygen, a WET-Lab FLNTU for turbidity (NTU) and Chlorophyll (ug/l) and, just during the second cruise, a Turner Cyclops sensor for the Chromophoric Dissolved Organic Matter (CDOM in ppb QS - Quinine Sulfate). The probe was calibrated on January 29th, 2013 at SBE Inc. in USA, but not before and after the two cruises, so data were just quality checked and processed by the SBE Data Processing™ software to correct the coarse errors.

All CTD data listed in table 1 are available in the SEANOE data repository (Ribotti et al., 2019a, b, c, and Ribotti et al., 2022) where each dataset is identified per cruise and provided as Ocean Data View (ODV, Schlitzer, 2022) ASCII file format with missing values set to -1.e$^{10}$.

Methodologies before, during and after any acquisition, instruments and personnel changed a little during the 21 years of CTD acquisitions, apart during the two coastal cruises SeriousGame2014-1 and -2 in 2014, due to different adopted instruments. This makes the data in the datasets highly coherent and comparable.

### 3. Data analyses

In 21 years, projects and scientific interests led to the choice of areas and sampling plans that could change in relation to vessel characteristics and sea conditions. Nevertheless, repetitions of casts were realized, like in the Sardinia Sea where the same sampling strategy was repeated for 7 cruises in 5 years or the sampling transects between Sardinia and Balears, across the Sicily Strait and the Corsica Channel and along the western Mediterranean, (see in table 1) mainly for climatological studies. In the following subsections, a process and a climatological event, observed from the data described in this paper, are mentioned to suggest further scientific research interests and opportunity for using these data.

In the first subsection we show step structures observed during the first four Medgoos cruises between 2000 and 2002 in the Sardinia Sea, but also during other cruises in the Western Mediterranean. In the following subsection we describe climatological studies, with a particular focus on the Western Mediterranean Transient, that was detected in the datasets acquired between 2005 and 2017.

### 2.1 Step structures in the deep Sardinia Sea

The first seven cruises named Medgoos have been organized covering the Sardinia Sea, partially or totally. Casts were realized on a regular 25 Km grid from the shelf to the abyssal plain, some hundreds of kilometres far from the coast. The integration of numerical models' solutions and satellite images permitted the study of the circulation in the Sardinia Sea (Ribotti et al., 2004; Puillat et al., 2006). It is mainly characterized by a fresher modified AW in the upper 200 m, a warmer and saltier Intermediate Water below and till 800 m depth moving northward along the Sardinian slope, and the

Western Mediterranean Deep Water between 800 m and the abyssal plain. Due to baroclinic instabilities of the Algerian
170    Current, surface and intermediate waters are dynamically interested by the presence of anticyclonic Algerian Eddies
(AEs) along the Sardinian coast, till a latitude around 40°N (Puillat et al., 2002; Pessini et al., 2020). Cruises Medgoos2,
4 and 7 (on March 2001, May 2002, and January 2004, respectively; see table 1) covered the whole Sardinia Sea
permitting to investigate some hydrological profiles of March 2001 and repeat some casts hundreds of kilometres far
from Sardinia. AEs were observed during the first two cruises in the southern-central part of the Sardinia Sea (Puillat et
al., 2006), while in January 2004 the area was eddy-free. The profiles from these three cruises showed step structures at
latitudes between 40.5° and 41.5°N, offshore north-western Sardinia at depths between 500 m and 1400 m. Following
the method proposed by Durante et al. (2019) to investigate the staircases systems in the central Tyrrhenian Sea, the
analysis of data collected during Medgoos cruises revealed always a 5 to 9 steps structure with interfaces of 50-110 m
thick located between intermediate and deep waters as depicted by the profiles of absolute salinity and conservative
temperature reported in figure 2.

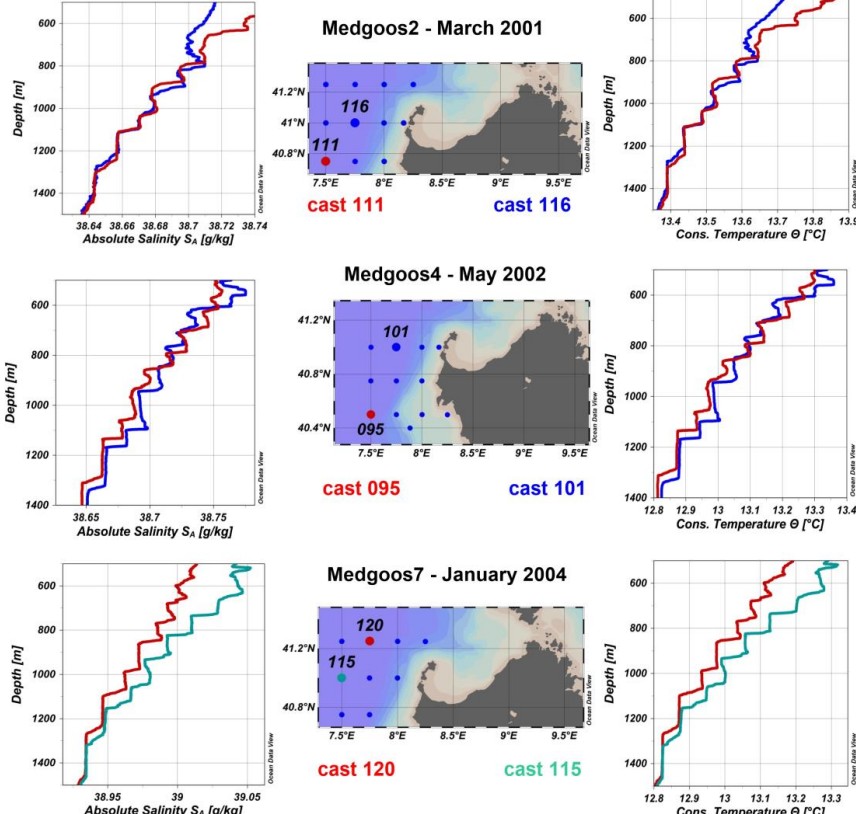

**Figure 2.** Some of the steps observed offshore north-west Sardinia between 500 and 1500 m through the profiles of absolute
salinity and conservative temperature at the interface of intermediate-deep waters in the period March 2001 - January 2004.

During the first cruise conducted in March 2001 the staircase structure was less evident than in 2002 and 2004. Each
step is defined by a layer with values of the stability ration ($R_P$) and of the Turner angle ($Tu$) finger regime, both defined
in Durante et al. (2019), comprised between 1 and 1.7 ($1 < R_P < 1.7$) and between 45° and 90° ($45° < Tu < 90°$),
respectively. Following Durante et al. (2019), these staircases systems can be defined with rough profiles, then
representing instabilities in the convective layer, and characterized by a reduced number of thick steps alternated by

rough interfaces. The ranges of the differences for temperature and salinity at the interface of each step ranged between

$1 \cdot 10^{-2}$ and $1.5 \cdot 10^{-1}$ °C and between $1 \cdot 10^{-2}$ and $9 \cdot 10^{-3}$ PSU, respectively, with an interface thickness varying between 9 m and 40 m. These values are comparable with those observed in the Tyrrhenian Sea by Zodiatis and Gasparini (1996) with the difference that the central Tyrrhenian Sea is a low energetic area in terms of water currents, whereas the northern Sardinia Sea is at the limits of the passage of deep anticyclonic eddies moving anti-clockwise in the Algerian sub-basin and generating intense currents (Pessini et al., 2018, 2020).

Staircases systems were also found in the central Algero-Provencal basin during the Bonifacio2011 (November 2011) and Ichnussa2012 (January 2012) cruises and were well identified from the analysis of the profiles of absolute salinity and conservative temperature profile as reported in figure 3.

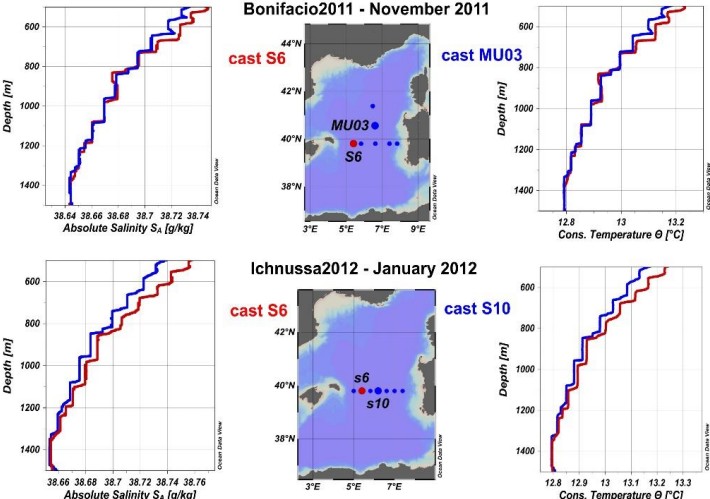

**Figure 3. Profiles of absolute salinity and conservative temperature of two casts where the steps were observed along the**
**transect between Balears and Sardinia at 500-1500 m depth in the period November 2011 – January 2012.**

These staircases are organized in 7 to 9 steps structures with interfaces 60-110 m thick and $R_P$ and $T_u$ comparable with those observed offshore north-west Sardinia. These systems were detected in sampling stations located in the central Provencal sub-basin and in the central part of the transect between Minorca Island in the Balears and Oristano in Sardinia, along latitude at around 40° N. They are similar to those described by Fuda et al. (2000) observed between the

500 and 1400 m depths from CTD and XBT profiles acquired in February 1994 in the open Algerian area.

The features of these dynamical structures confirm they are related to salt fingering originated due to the presence of intermediate waters, both when close to the continental slopes (NW Sardinia), or when in the middle of the Algero-Provencal area. In both cases, their presence is dependent on the absence of eddies.

Finally, during 12 cruises between March 2003 (Medgoos6) and December 2015 (Ichnussa2015), the 3550 m deep cast

number 51 in the central Tyrrhenian Sea was repeated to study the staircase system previously mentioned and recently well described by Durante et al. (2019).

### 2.2 Climatology of the western Mediterranean sub-basin

In the Mediterranean the deep circulation is organized in two deep cells, located in the western and eastern sub-basin respectively, and constituted by denser waters than those above whose thermohaline features can differ from between



the two sub-basins (Malanotte-Rizzoli et al. 1997; Sparnocchia et al., 1999; Wu et al., 2000; Bethoux et al., 2022; Rixen
      et al., 2005; Marty and Chiavérini, 2010; Bensi et al., 2013; Ingrosso et al., 2017; Send and Testor, 2017).

      Between October 2004 (Medgoos9) and 2017 (Ichnussa2017), fifteen cruises covered some deep sub-basins of the
      western and central Mediterranean, from the Sicily Channel and the Tyrrhenian Sea over the Gibraltar Strait, with
      hydrological casts (see table 1). These data, apart the shallow cruise Bonifacio2010-Cor and the two SeriousGame in

2014, permitted to follow the spreading of the new Western Mediterranean Deep Waters (nWMDW) in the western
      Mediterranean sub-basins since its formation in 2005 (Schroeder et al., 2008), revealing over a decade of great
      thermohaline variability. During winter 2004/2005, above the north-western Mediterranean area, a little rain with
      intense and persistent northerly winds brought a massive heat loss in the Gulf of Lions, the highest since 1948, that led
      to an increase in surface salinity (Font et al., 2007). In the same years, the propagation of the Eastern Mediterranean

Transient, from the eastern to the western basin of the Mediterranean, led to maximum values of temperature and
      salinity in the intermediate layers (Gasparini et al., 2005; Lopez-Jurado et al., 2005; Schroeder et al., 2006; Margirier et
      al., 2020). The concomitance of these meteorological and oceanographic processes has created the perfect conditions
      for the formation of warmer and saltier deep waters than the pre-existing ones (Herrmann et al., 2010; Lopez-Jurado et
      al., 2005), then occupying the deepest layer close to the bottom and known as Western Mediterranean Transient (WMT;

Lopez-Jurado et al., 2005; Schroeder et al., 2008; Briand, 2009; Zunino et al., 2012). After the first event of 2004/05,
      WMT continued with large volumes of nWMDW formed in winter 2005/06 (Medgoos10&13 in figure 4) while the
      following 2006/07 and 2007/08 no deep water production occurred due to a mild "cold" season. But deep convection
      started again in winter 2008/09 and in the following (i.e. 2009/10, 2010/11 and 2011/12) due to particularly cold
      winters, like 2011/12, in the Mediterranean Sea.

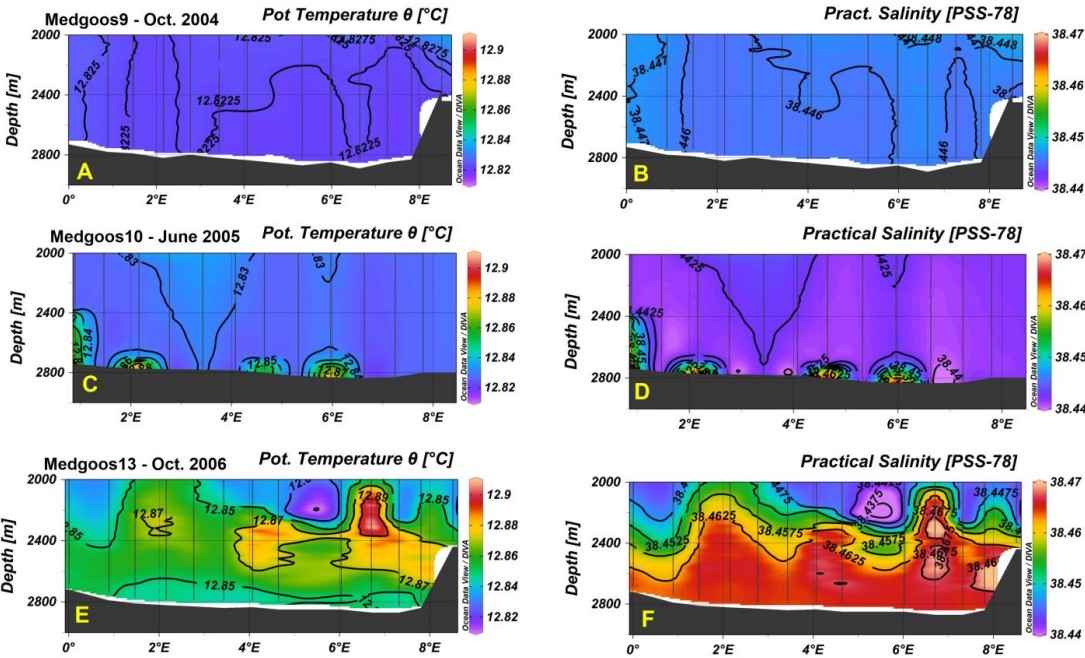


**Figure 4. Deep sections (over 2000 m depth) of potential temperature (°C) and practical salinity (dimensionless) along a transect      at 37-38° N between the eastern entrance to the Alboran Sea (0°E) and the western entrance to the Sardinia**

Open Access    Earth System
**Channel (8.5°E) during the cruises Medgoos9 (A-B; October 2004), before the WMT, Medgoos10 (C-D; May-June 2005) and Medgoos13 (E-F; October 2006) with the filling of the Algerian sub-basin by the WMT.**

Every winter, new warmer, saltier, and denser deep waters formed, moving to a continuous increase in heat and salt content in the deep layer, making the deep temperature-salinity (θS) diagrams more and more complex (Schroeder et al., 2016). The replacement of the old deep waters near the bottom by new ones, is visible as a creep close to the bottom in 2005, becoming 600 m thick in 2006, almost 1000 m thick in 2008, over 1200 m in 2010, 1400 m in 2013 and > 1500 m in 2015. Since 2010 the signal of the WMT was found also inside the Tyrrhenian Sea, after having overpassed the

Sardinian Trough, connecting the Algerian sub-basin with the Tyrrhenian one along the axis of the Sardinia Channel. The WMT changed structure and properties of the deep layers in the Western Mediterranean recognizable by hooks and inversions in characteristic θS diagrams (in figure 5 in comparison with Medgoos9 data, prior the WMT). This set of CTDs acquired during over a decade (2004–2017) can contribute to the understanding of the temporal and spatial evolution of the thermohaline variability in the western Mediterranean Sea. The thermohaline circulation of the

Mediterranean Sea, through the Mediterranean Outflow Water (MOW), has a particular impact on the North Atlantic circulation and the Atlantic Meridional Overturning Circulation (AMOC) due to its high salinity, as rebuilt in palaeoceanographic studies (Rogerson et al., 2012; van Dijk et al., 2018; Ausín et al., 2020; Mesa-Fernández et al., 2022).

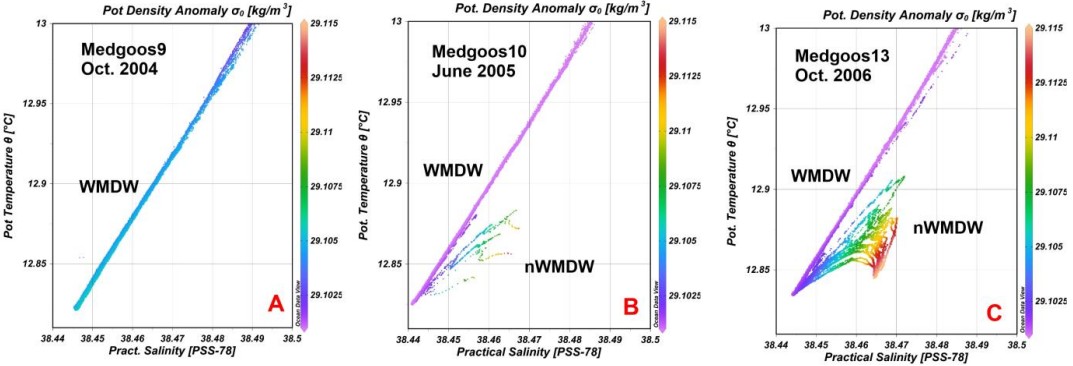

**Figure 5. Potential temperature – practical salinity (ϴS), with potential density anomaly ($\sigma_\theta$; Kg/m3) of the casts along the transect of figure 4 during the cruises Medgoos9 (A; October 2004), before the WMT, Medgoos10 (B; May-June 2005) and Medgoos13 (C; October 2006) with the typical hooks and inversions of the new deep waters' characteristics.**

## 4. Data availability

The four datasets described in this study are publicly available and free of charge from the SEANOE data repository as

Ribotti et al. (2019a, b, c) in https://doi.org/10.17882/59867, https://doi.org/10.17882/70340 and https://doi.org/10.17882/59777 respectively, and as Ribotti et al. (2022) in https://doi.org/10.17882/87567. The presented datasets are composed of CTD data in Ocean Data View (ODV) TXT Spreadsheet files, one for each cruise.

## 5. Discussion and conclusion

Over 1468 hydrological vertical profiles were acquired during 29 oceanographic cruises in 21 years in the

Mediterranean Sea, with a particular focus on the western and central Mediterranean sub-basins, between May 2000 and October 2020. Each cruise was mainly led by the projects' objectives funding the research but also by scientific





interests at different spatial (coastal or offshore) and temporal (from days to years) scales. Use and reuse of data with time was mainly driven by the following issues: solution of marine environmental problems, climatology, and/or model validation. This is visible in the different spatial resolution of the sampling plans, on a regular grid of casts or along repeated transects, sometimes both used during the same cruise.

During the years and despite of some improvements in sensors quality, sensors and instruments were prepared through the same procedures before and, after their use, the same quality assurance protocols, apart in the two Serious Games cruises in 2014. CTDs data followed all quality check, assessment, standard and best practices, defined at international level (see Bushnell et al., 2019) and through standardized procedures for all sensors (Hood et al., 2010). Pre- and post-calibration, redundant sensors, on-board analyzed water samples has achieved the best accuracy standard for the used instruments. The presence of the same operators on-board in most of the years has then assured a reduced uncertainty of measurements further increasing the quality of the data set.

The final results are coherent datasets of CTD data that can be used for all studies, as described in the paper for climatology or for salt fingering.

**Author contribution**

AR led projects funding data acquisition and cruises, finalized data quality procedures described in the paper, realised datasets and the writing of the paper. MB led cruises, finalized data quality procedures described in the paper, realised datasets and collaborated to the paper writing. RS and AC led projects funding data acquisition and collaborated to the data analyses and paper writing. FP and QC collaborated to the data analyses and paper writing.

**Competing interests**

The authors declare that they have no conflict of interests.

**Acknowledgments**

The data used in this work have been collected in the framework of several national and European projects, i.e. the Italian MIUR project SIGLA (Sistema Integrato per la Gestione delle Lagune e dell'Ambiente marino costiero), the Italian ASI project PRIMI (Progetto Pilota Inquinamento Marino da Idrocarburi), the EU projects MyOcean (contract 218812) and MyOcean2 (contract 283367), the MED programme project MEDESS4MS (Mediterranean Decision Support System for Marine Safety; agreement MED2S-MED11-01), the Italian MATTM project SOS-BONIFACIO (prot. DPN-2009-0001027 of 20/01/2009), the Italian MIUR project PON TESSA (Sviluppo di TEcnologie per la 'Situational Sea Awareness', agreement PON01_02823), the Italian MIUR project RITMARE (La Ricerca ITaliana per il MARE), the EU COMMON SENSE contract n. 614155), the Italian MATTM project SOS-Piattaforme & Impatti offshore (Reg. Uff. U. 0000939.17-01-2017), 2014 - 2020 INTERREG V-A Italy - France (Maritime) project SICOMAR plus (Transborder System for Safety at Sea against the Risks of Navigation and for the Protection of the Marine Environment plus). We thank captains and crews on R/V Urania, R/V Maria Grazia, R/V Minerva Uno, CP-406 Antonio Scialoja and R/V G. Dallaporta for their essential support on board, then Dr Paschini Ezio and Mr Penna Pierluigi from CNR in Ancona for their support during the cruises SeriousGame2014. Finally, we thank Dr. Mario Astraldi and Dr. Salvatore Mazzola (deceased in January 2021), two great Italian oceanographers for their support and suggestions in the first ten years of our work at sea.



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
