# Peer review of "Twenty-one years of hydrological data acquisition in the Mediterranean Sea: quality, availability, and research"

_Earth System Science Data, 2022_

## Referee Comment (RC1)

The manuscript is well and clearly written and is fluent and easy to read. The content and data fit the objectives of the ESSD journal. For these reasons, I support its publication. I would like to add just a couple of comments that may help to clarify the text:

Line 206: substitute "due to" with "by";

Line 208: Explain why you expect to find staircases away from the eddies, or why you don't expect to find them in the eddies;

Line 220: add in the text the typical thermohaline properties of the nWMDW;

Figure 4: add the track of these casts.

---

## Editor Comment (EC1)

[revised manuscript text omitted]

---

## Author Comment (AC2)

To
Dr. Giuseppe M.R. Manzella
ESSD Guest Editor
and
the ESSD Editorial Support Team

Oristano, 16/08/2022

Subject: answer to Editor's and Reviewers' comments on "Twenty-one years of hydrological data acquisition in the Mediterranean Sea: quality, availability, and research" by Alberto Ribotti et al., Earth Syst. Sci. Data Discuss., https://doi.org/10.5194/essd-2022-168-RC1, 2022

Dear Editor and Reviewers,

thank you very much for your suggestions that we, as authors, have always considered strongly valid and which have helped us to improve the level of our manuscript more and more.

Here the specific answers to both Reviewers and Editor requests follow, in red for a better readability. About Editor's suggestions, only the second group of comments will be shown here, even if the manuscript also includes changes due to his first comments received after our submission.

Best regards,
* * *
**Anonymous Referee #1**
**Referee comments**

The manuscript is well and clearly written and is fluent and easy to read. The content and data fit the objectives of the ESSD journal. For these reasons, I support its publication. I

would like to add just a couple of comments that may help to clarify the text:

Line 206: substitute "due to" with "by";                                        done

Line 208: Explain why you expect to find staircases away from the eddies, or why you don't expect to find them in the eddies;                                        done then adding a reference. For a complete overview, we also mentioned the Sardinian Eddies in addition to the Algerian Eddies with two further references

Line 220: add in the text the typical thermohaline properties of the nWMDW;   added with the corresponding bibliographic reference

Figure 4: add the track of these casts.                                        done

**Anonymous Referee #2**
**Referee comments**

The manuscript is clearly written, easy to read and presents very important and interesting datasets for the scientific community. The content and data fit the objectives of the ESSD journal. The data availability is clearly defined in the manuscript. For all the above I propose that the article should be published in its present form. I am only proposing some very minor corrections:

The numbering of the subsection is wrong and needs correction

line 18: Remove "Then",                                                       done

line 149: "apart from",                                                       done

line 170: change "dynamically interested" by "dynamically affected",          done

line 193: change "energetic" by "energy"                                      done

**Editor**
**2ⁿᵈ round of comments**

line 29: May be the entire CNR organised more than 29 cruises in the Mediterranean. This statement is probably valid only for one or two institutes or groups          corrected

line 38: climatological studies or climate variability/change          changed in "studies on hydrological variability"

line 50: add: paragraph 2 ...                                    added paragraphs 2 and, below, 3

line 67: it is not clear. Do you mean that data collection outside the projects' aims were limited? May you provide an example?                          this sentence was deleted

line 150: Delayed mode data collected during different cruises need a complex quality check that is importante to be described in a paragraph. The authors should give answers to many questions; - climatology/climate: have you checked the values below the LIW? Are there changes? - have the data been compared with existing climatologies8e,g, SeaDataNet or CMEMS reanalysis)? - have data flagged? How? and if not, why?                          The Editor is right, but these data have been controlled following procedures developed, tested, improved, and used for years in almost all listed cruises and by other CNR institutes, as described in this paragraph. So, none of the above mentioned steps was followed in our quality check during data preparation for the SEANOE database. What the authors improperly did was the use of the word "climatological studies" or "climatology" along the text that was changed with "oceanographic studies" or "variability" or other more correct terms.